# AMS-Quant: Adaptive Mantissa Sharing for Floating-Point Quantization

## Abstract

Large language models (LLMs) have demonstrated remarkable capabilities in various kinds of tasks, while the billion or even trillion parameters bring storage and efficiency bottlenecks for inference. Quantization, particularly floating-point quantization, is known to be capable of speeding up LLM inference by reducing memory footprint and data movement during the inference process. For the first time, we advance the floating-point quantization exploration from integer bit-widths to non-integer bit-widths, namely **AMS-Quant**, to further approach the quantization sweet spot. AMS-Quant incorporates two novel techniques to put it into effect: (1) it proposes Mantissa-bit Sharing, which groups k quantized weights and lets them share the least significant mantissa bit, allowing us to further approach the minimum quantization bit-width without accuracy loss. (2) It introduces Adaptive Searching, which employs an offline optimization strategy to minimize the accuracy degradation introduced by sharing. Moreover, AMS-Quant is also prototyped as efficient CUDA Linear kernels, which translates memory savings into wall-clock latency reduction by reducing memory access. Extensive experiments on large-scale datasets and models show that AMS-Quant can quantize the model to FP-5.33-e2m3 and FP4.25-e2m2, and significantly speed up the LLM decoding over FP16 inference ($2.8\times$ and $3.2\times$), with negligible accuracy loss.

## 1 Introduction

Large language models (LLMs) have demonstrated remarkable capabilities in reasoning, code generation, and complex problem-solving. This power, however, comes at the cost of enormous storage and computational demand. For example, the recently released Deepseek-V3/R1 (DeepSeek-AI, 2024; 2025) contains 671B parameters and requires server-scale clusters to run. Even models considered "medium-scale", such as Llama-3.1-8B (AI@Meta, 2024), demand more than 16GB of memory just to store weights in FP16 precision. These storage and efficiency bottlenecks make compression indispensable for practical deployment.

Quantization is one of the most direct compression techniques: by representing weights in reduced-bit-width formats, it simultaneously shrinks memory footprint and speeds up memory-bound inference. The challenge, however, is that aggressive bit-width reduction usually results in a non-negligible accuracy drop. Hence, a central problem in quantization research is to identify the *best trade-off* between efficiency and accuracy.

Early studies focused on INT quantization (Xiao et al., 2023; Frantar et al., 2023; Lin et al., 2024), for it was better supported in AI hardware. Later, researchers observed that **floating-point quantization** better matches the bell-shaped weight distribution of LLMs, as shown in Figure 2, leading to less accuracy degradation at the same bit-width (Kuzmin et al., 2022; Si et al., 2025). FP6-LLM (Wu et al., 2023; Xia et al., 2024) shows that FP6 (e3m2) retains nearly the same accuracy level as FP16 across various LLMs, such as GPT and Llama families. FPE2M2 (Yi et al., 2025) further shows that FP5-E2M2 largely outperforms INT5 on models up to 13B. Nevertheless, when pushing bit-width further to 4 bits, both FP4 and INT4 suffer from severe model accuracy loss, though FP4 demonstrates less quality compromise. Collectively, these observations suggest that floating-point formats consistently outperform integer formats in low-bit regimes, and that the quantization "sweet spot", which optimally trades off between efficiency and accuracy, likely lies in FP6 or FP5.

Figure 1: Illustration of AMS-Quant Mantissa sharing. For the sake of simplicity, we take FP4.25 as an example.

In this work, we ask a natural but overlooked question: *must the sweet spot be an integer bit-width?* To answer this question, we propose **AMS-Quant**, a new method that realizes non-integer bit-width floating-point quantization. It consists of two ideas: 1) *Mantissa-bit Sharing*: we group $k$ quantized weights and let them share the least significant mantissa bit. This advances our investigation of quantization from integer bit-widths FPx to non-integer bit-widths FP(x-1).y where y $= 1/k$, allowing us to further approach the quantization sweet spot; 2) *Adaptive Searching*: we employ an offline optimization strategy to minimize the accuracy degradation introduced by sharing. Moreover, we also implement efficient CUDA Linear kernels that restore AMS-Quantized weights during runtime, which translates memory savings into wall-clock latency reduction by reducing memory access.

Our experiments of AMS-Quant on various models and datasets manifest its effectiveness. For example, AMS FP5.3-e2m3 quantization demonstrates the same accuracy level as FP16, while reducing 66.7% storage/memory requirement, and achieves up to 2.77x acceleration over FP16 in certain memory-bound scenarios.

In summary, the contributions of this work are as follows:

- We proposed **AMS-Quant**, a novel quantization method that realizes non-integer bit-width quantization, in order to advance quantization exploration from integer bit-widths to non-integer bit-widths, and further approach the quantization sweet spot.

- We devised 2 key strategies of AMS-Quant, *Mantissa-bit sharing* and *Adaptive searching* mechanism, which powers non-integer bit-width quantization and ensures minimal information loss.

- Coped with the carefully designed GPU kernels, AMS-Quant brings 2.77x inference acceleration for FP-5.33-e2m3 quantization with the same accuracy level as FP16, and a sweet sport FP-4.25-e2m2 quantization with 3.2x acceleration and negligible accuracy drop.

| | E2M3 | E3M2 |
|---|---|---|
| **Exponent Bias** | 1 | 3 |
| **Infinities** | N/A | N/A |
| **NaN** | N/A | N/A |
| **Max Normal** | S 111 $11_2 = \pm 2^2 \times 1.875 = \pm 7.5$ | S 111 $11_2 = \pm 2^4 \times 1.75 = \pm 28.0$ |
| **Min Normal** | S 001 $00_2 = \pm 2^0 \times 1.0 = \pm 1.0$ | S 001 $00_2 = \pm 2^{-2} \times 1.0 = \pm 0.25$ |
| **Max Subnormal** | S 000 $11_2 = \pm 2^{-1} \times 0.875 = \pm 0.875$ | S 000 $11_2 = \pm 2^{-2} \times 0.75 = \pm 0.1875$ |
| **Min Subnormal** | S 000 $01_2 = \pm 2^{-1} \times 0.125 = \pm 0.125$ | S 000 $01_2 = \pm 2^{-2} \times 0.25 = \pm 0.0625$ |

Table 1: Comparison of E2M3 and E3M2 Floating Point Formats (OCP, 2023)

## 2 BACKGROUND

### 2.1 QUANTIZATION

**Round To Nearest Quantization**

Following a general definition (Kuzmin et al., 2022), we define the quantization operation $Q(W)$ as

$$Q(W) = \text{Round}(\frac{W}{s_q}), \quad s_q = \frac{\max(|W|)}{M} \tag{1}$$

For integer quantization, the `Round()` function is the usually understood round-to-nearest integer function, but for floating-point quantization, it means round-to-nearest floating-point number within the chosen data format. Without loss of generality, we define the round-to-nearest operation as:

$$\text{Round}(w) = \arg\min_{\alpha} |w - \alpha|$$

where $\alpha \in \{\alpha_1, ..., \alpha_n\}$ is the set of all possible values of the given data format, whether it is integer or floating-point, and the dequantization reverses the quantization operation by multiplying the quantized weights with their quantization scales:

$$DeQ(W) = Q(W) \times s_q \tag{2}$$

However, the rounding operation is irreversible, which is the source of quantization error and model capability drops.

**Weight-only quantization** Depending on how to quantize the activations, quantization methods can be classified as *weight-activation* and *weight-only* quantization. As weight-only quantization preserves the bit-width activations as FP16, the quantized weights must be dequantized before they can be multiplied. Hence, unlike weight-activation quantization, weight-only quantization cannot speed up the actual computation process. Instead, it mainly accelerates the memory access process, which accounts for the majority of latency during the token generation (decoding) stage and other memory-bound scenarios.

### 2.2 FLOATING-POINT FORMATS

The IEEE-754 (IEEE, 2019) standard defines the widely-adopted floating-point number system as follows:

$$x = \begin{cases} (-1)^S * 2^{(E-bias)} * (1 + \frac{d_1}{2^1} + \frac{d_2}{2^2} + \cdots + \frac{d_m}{2^m}) & E \neq 0 \\ (-1)^S * 2^{(1-bias)} * (\frac{d_1}{2^1} + \frac{d_2}{2^2} + \cdots + \frac{d_m}{2^m}) & E = 0 \end{cases}$$

where $bias = 2^e - 1$. There are also special values defined in IEEE-754 when the exponent bits are all "1"s, defined as Infinity or NaN (Not a Number). For AMS-Quant, as we will ultimately perform dequantization for the quantized weights to FP16 (§3.2), the exponent bits would never be full "1"s. Hence, under AMS-Quant floating-point system, the special values of infinity and

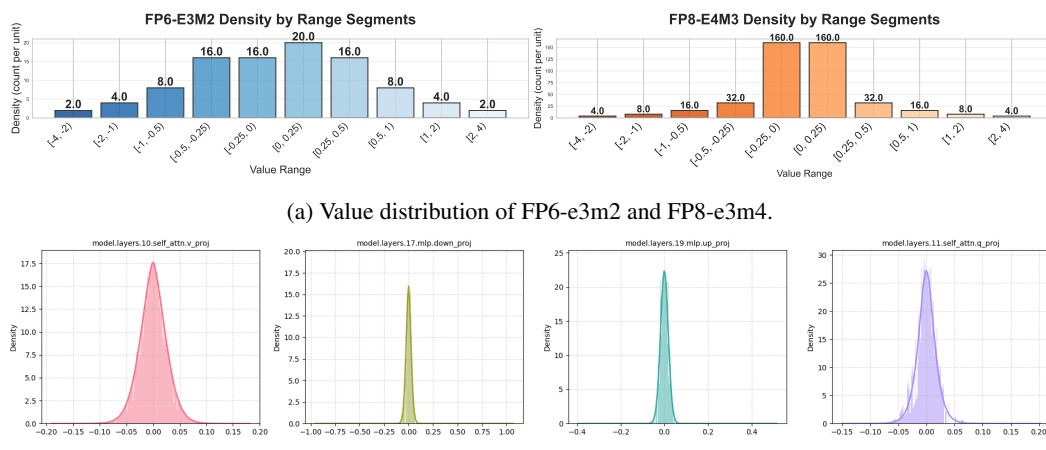

(a) Value distribution of FP6-e3m2 and FP8-e3m4.

(b) Model weights distribution across different layers randomly taken from four models: Qwen3-4B-Instruct-2507, Qwen3-8B, Llama-3.2-3B-Instruct, and Llama-3.1-8B-Instruct.

Figure 2: Figure 2a shows the value distribution of FP formats, and Figure 2b shows the weight distribution of 4 randomly chosen layers from different models. All follow bell-shaped distributions.

Nan will be replaced by regular values calculated using the above equations. This aligns with the MicroScaling Quantization format (Rouhani et al., 2023). Table 1 shows an example FP6 scheme.

The distribution of the Floating-Point number system appears to be a bell-shaped curve, which naturally fits the LLM weights distribution, as shown in Figure 2, hence incurs less information loss compared with INT quantization.

## 2.3 QUANTIZATION SWEET SPOT

Quantizing an LLM to a lower bit level leads to higher inference efficiency yet often higher capability loss as well. However, the relationship between the efficiency gain and capability loss is not linear. Usually, quantizing a 32-bit float32 model to 16-bit float16 can ideally speed up 2x inference but results in no performance drop, and a 16-bit float16 model to FP8 weights or INT8 weights further accelerates 2x inference, and starts to witness little performance degradation. It is when we continue quantizing the 8-bit model to lower bits that we start to identify some statistically significant accuracy drops in benchmark datasets, but the loss is so small that it is worth doing this quantization. Researchers and engineers strive to find the best bit level for LLMs that can best trade off efficiency and accuracy, which is the sweet spot that leads to the most efficiency gain but the least accuracy drop. Arguably, while different models should have different sweet spots, we believe that the most widely used LLM models within similar scales share similar sweet spots.

Quant-LLM (Xia et al., 2024; Wu et al., 2023) claimed that FP6 is the "sweet spot" of floating-point RTN quantization. However, they only examined the e3m2 scheme of FP6 and benchmarking it against int4 quantized models, ignoring e2m3 scheme and the FP5. Later, FPE2M2 Yi et al. (2025) pointed out that "E2MX consistently dominates under different bit width" under the assumption that LLM weights obey a Gaussian Distribution, and it is FP5-e2m2 that should be the optimal choice.

We conducted a preliminary study using the Llama-3.2-3B-Instruct and the latest Qwen3-4B-Instruct-2507. We apply the naive RTN quantization with respect to different schemes, and evaluate the accuracy of these quantized models on GSM8k (Cobbe et al., 2021). Results are plotted in Figure 3. While the accuracy of FP4-e2m1 drops significantly, FP5-e2m2 achieved little loss, less than 2%. Nevertheless, FP6-e2m3 gets closer to the full-precision model, with an accuracy drop of less than 0.5%. We also noted that the FP6-e3m2 get similar or even the same accuracy results as FP5-e2m2, meaning that the extra bit used in exponent is not made the most use as it is used in mantissa, indicating that the dynamic range of these LLM weights could be covered using 2-bit exponents, and the more mantissa bits can help mitigate model quality loss due to quantization error (Kuzmin et al., 2022; Yi et al., 2025). Based on the preliminary study, we argue that the actual quantization sweet spot lies in the open interval between FP6-e2m3 and FP4-e2m1.

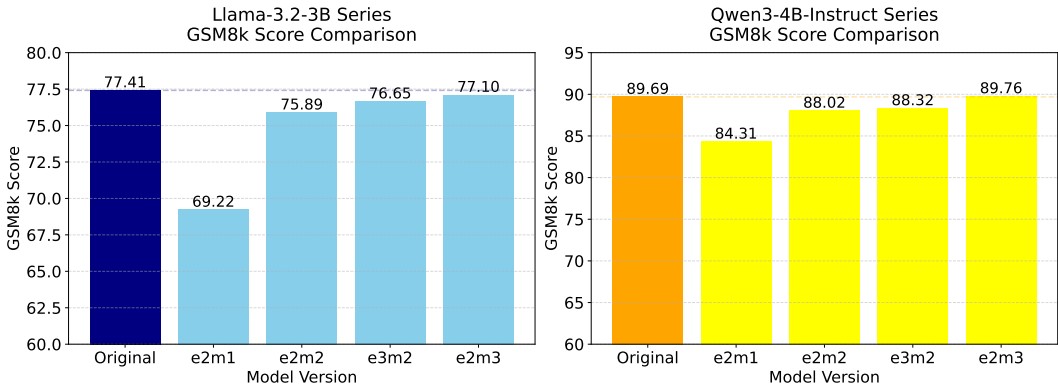

Figure 3: Different RTN quantized models of Llama-3.2-3B-Instruct and Qwen3-4B-Instruct-2507 on GSM8k. FP6-e2m3 could retain the same accuracy level as the full precision of FP16, while FP6-e3m2 and FP5-e2m2 incur little performance degradation,and the FP4-e2m1 suffers from a serious loss.

## 3 AMS-QUANT

### 3.1 ADAPTIVE MANTISSA SHARING QUANTIZATION

As shown in Figure 1, we present a novel approach called **Adaptive Mantissa Sharing Quantization (AMS-Quant)**, which converts original model weights from FP16 to FPx, and then lets $k$ quantized FPx weights share their last mantissa value with the same bit, leading to FP(x-1).y quantization where y=$1/k$. The method combines several techniques to improve the trade-off between quantization efficiency and model accuracy. Specifically, we employ **Channel-wise Quantization**, **Mantissa Sharing**, and **Adaptive Searching** to achieve a compressed weight representation that retains significant accuracy.

**Channel-wise Quantization**  The first step in AMS-Quant involves applying *Round-to-Nearest (RTN) quantization* to the original FP16 weights, converting them to a lower-precision floating-point format, denoted as FPx. This quantization is performed channel-wise. By doing so, we inherently generate a set of quantized weights that follow a logarithmic distribution, which is proven to be more aligned to the original distribution of LLM weights than INT-quantized weights, preserving the model's accuracy while reducing the number of bits required to represent each weight.

**Mantissa Sharing**  Once the weights are quantized to FPx, we perform *Grouped Mantissa Sharing*. This step groups the quantized weights based on a parameter $k$, which determines the number of weights to share the same least significant mantissa bit. Importantly, we apply mantissa sharing along the *input-channel* dimension of the weight tensor. This choice is motivated by the observation that activation outliers typically exhibit a channel-wise pattern. By aligning the grouping dimension with the channel-wise structure of activation outliers, we mitigate the sensitivity of shared mantissas to such outliers, thereby reducing the potential accuracy degradation introduced by group sharing.

**Adaptive Searching**  The final step in AMS-Quant is *Adaptive Searching*, where we refine the shared mantissa bit. For each group of quantized weights, we explore all possible values for the last mantissa bit. For each candidate value, we choose the one that minimizes the *Mean Squared Error (MSE)* between the restored weights and the original model weights. This searching process is briefly described as the following formula, where $DeQ$ is the dequantization operation defined in Eqn and 2 and $G(FPx_i, m_0)$ represents the operation to set the last bit of $FPx_i$ to $m_0$

$$m_0^* = \min_{m_0 \in \{0,1\}} \sum_{i=1}^{3} (deQ(G(FP6_i, m_0)) - FP16_i)^2$$

By carefully selecting the last mantissa bit for each group, we can balance information loss and model performance. This adaptive method significantly reduces quantization error, ensuring that the final representation of the weights closely matches the original FPx weights.

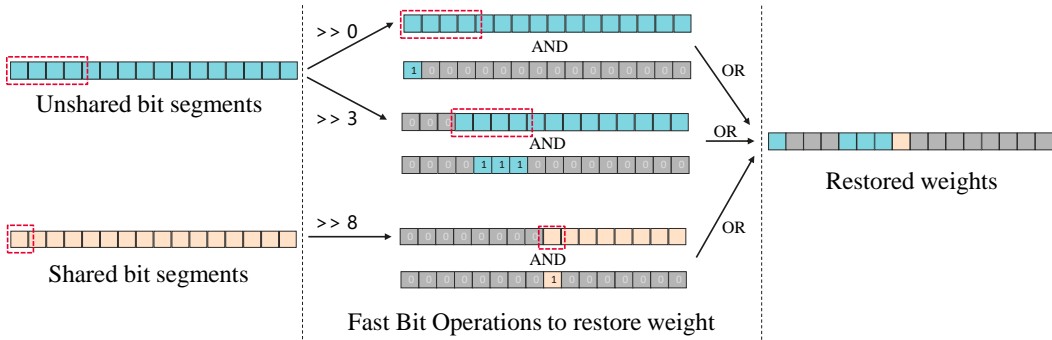

Figure 4: Illustrative example of restoring a prepacked AMS-Quant data back to FP16 via bit-level operations.

## 3.2 FAST RESTORATION VIA BIT OPERATIONS

Modern large language models (LLMs) are typically deployed on dedicated AI accelerators (*e.g.*, NVIDIA GPUs, Intel XPUs). As with other computing devices, these accelerators must load model weights and input data from memory into compute cores to perform the actual computation. Contemporary accelerator architectures generally favor memory accesses aligned to *regular bit-widths*, i.e., powers of two such as 16-bit or 32-bit, since irregular bit-widths often require padding or additional preprocessing, thereby reducing end-to-end inference efficiency.

Inspired by the TC-FPx framework (Xia et al., 2024), we design efficient memory access schemes for irregular bit-width quantizations in the FPx.y family by prepacking multiple quantized weights into standard data types. For example, FP6 weights can be split into two portions, a 4-bit high segment and a 2-bit low segment (the $(4 + 2)$ scheme). Across 16 weights, the high segments are stored in four `uint16` words, while the low segments are stored in two `uint16` words, requiring six memory accesses in total. Analogous layouts are adopted for other FPx.y formats with AMS quantization, where $k$ weights share the least significant mantissa (LSB) bit. Taking FP4.25-e2m2 as an example, four FP5-e2m2 weights share one mantissa bit. Thus, we can pack $16 \times 4 = 64$ quantized weights into one `uint16` word for the shared LSBs and 16 `uint16` words for the remaining 4-bit segments. This layout enables efficient bulk loading of quantized weights, followed by register-level reconstruction. After loading, each packed word undergoes bit-level *SHIFT*, *AND*, and *OR* operations to restore the sign, exponent, mantissa, and shared mantissa bits, yielding FP16 weights.

## 3.3 KERNEL ROADMAP

Building upon the TC-FPx framework, we design SIMT-friendly kernels that support our quantization methods. The kernel workflow consists of three main stages:

1. **Ahead-of-time weight packing.** FP(x-1).y weights are prepacked before runtime. For segmented formats (e.g., FP4.25), weights are stored in multiple segments (e.g., 4 high bits and 1 LSB for FP4.25). This packing method applies to most formats, except for one special case where FP5.33 allows three weights, along with a shared LSB, to fit neatly into one half-word, enabling continuous packing without segmentation.

2. **Weight unpacking (runtime).** At runtime, segmented weights are reconstructed by stitching their segments into the target FPx representation.

3. **Thread-level dequantization.** At runtime, the stitched (or directly loaded) weights are dequantized in parallel at the thread level, producing FP16 values ready for tensor core MMA operations.

Overall, our unified bit-level restoration framework improves memory-bound LLM inference by maximizing memory bandwidth utilization and reducing bandwidth pressure. Prepacking multiple quantized weights into standard word sizes enables coalesced memory loads, while segment stitching and thread-level dequantization reconstruct multiple FP16 values per thread without increasing

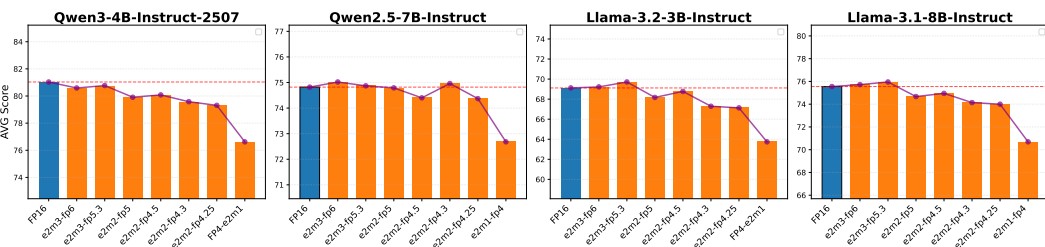

Figure 5: Average scores of accuracy benchmark results for the 4 chosen models. We plotted the results in the order of decreasing bit-width to visualize the trend of accuracy changes(drops). For all the tested models, FP5.3 consistently shows the same accuracy level as FP16. The sharpest turning point appears at FP4.3 or FP4.25, **indicating the quantization sweet spot.**

arithmetic operations. In FP5.33, the natural alignment to half-precision words further minimizes unpacking overhead. By focusing on efficient memory transfer and parallel restoration, the framework ensures that memory-bound operations(e.g. GEMV linear layers) can saturate the memory subsystem, leading to high throughput and low-latency inference across arbitrary FP(x-1).y formats.

## 4 EXPERIMENTS

We conducted experiments to evaluate our AMS-Quant on both accuracy and efficiency aspects.

### 4.1 ACCURACY

**Models** For models, we chose Llama-3.2-3B-Instruct and Llama-3.1-8B-Instruct from the Llama family (AI@Meta, 2024), and Qwen3-4B-Instruct-2507 and Qwen2.5-7B-Instruct from Qwen family (Team, 2025; 2024). We did not choose Qwen3-8B as it was a hybrid model tuned with thinking and non-thinking data, hence may show inconsistent behaviors from other models.

**Tasks and datasets** We used OpenCompass [1], a widely used LLM evaluation suite, to conduct our experiments. For benchmark datasets, we chose MMLU(`mmlu_gen_23a9a9`) (Hendrycks et al., 2021) for general language understanding and knowledge testing, IFEval(`IFEval_gen_3321a3`) (Zhou et al., 2023) for instruction following, and GSM8k(`gsm8k_gen_17d0dc`) (Cobbe et al., 2021) for general reasoning abilities. There are several metrics available in each task. We reported stricter versions of these metrics, and thus "Accuracy" for MMLU and GSM8k, and "Prompt-level-strict-accuracy" for IFEval.

**Comparison** We apply AMS-Quant to the above chosen models, and compare our results with the other vanilla RTN quantized schemes, including the *de facto* inference format FP16-e5m10, and other possible low-bit floating-point quantization schemes such as FP4-e2m1, FP5-E2M2 (Yi et al., 2025), FP6-e3m2 (Xia et al., 2024), and FP6-e2m3. Our AMS-Quantized models involved FP5.3-e2m3, FP4.5-e2m2, FP4.3-e2m2, and FP4.25-e2m2.

**Main Results** The main accuracy results are summarized in Table 2 and Figure 5. Generally speaking, FP6-e2m3 quantized models retain the same accuracy level as the original full FP16 model, while FP5-e2m2 models lose around 1∼2%, and FP4-e2m1 models suffer from 5∼10% degradation. Nevertheless, the AMS-Quantized models demonstrate good performance. Even FP4.25-e2m2 can reach a similar accuracy level as FP5-e2m2, far better than FP4-e2m1. As a result, we argue that FP4.25 is closer than FP5 to the quantization sweet spot of these models that optimally balances model quality and size reduction. What's more, we found that FP5.3-e2m3 consistently gets close accuracy to FP6-e2m3, yet achieved a higher compression ratio, making FP5.3 a better quantization bit-width compared with FP6 and FP5, in terms of both model accuracy and inference efficiency.

---

[1]`https://github.com/open-compass/opencompass/tree/0.4.2`

Table 2: Accuracy results of different quantized models on the IFEval, GSM8k, and MMLU benchmarks. We highlight the best results with **bold** fonts.

| Model | IFEval | GSM8k | MMLU | Avg. |
|---|---|---|---|---|
| **Qwen3-4B-Instruct-2507** | | | | |
| FP16 | 80.96 | 89.69 | 72.45 | **81.03** |
| FP6 (e2m3) | 79.85 | 89.76 | 72.17 | 80.59 |
| FP5.3 (e2m3) | 81.70 | 88.86 | 71.74 | 80.77 |
| FP5 (e2m2) | 81.52 | 88.02 | 70.18 | 79.91 |
| FP4.5 (e2m2) | 80.04 | 88.70 | 71.49 | 80.08 |
| FP4.3 (e2m2) | 78.19 | 89.76 | 70.79 | 79.58 |
| FP4.25 (e2m2) | 79.30 | 88.32 | 70.30 | 79.31 |
| FP4 (e2m1) | 77.82 | 84.31 | 67.72 | 76.62 |
| **Qwen2.5-7B-Instruct** | | | | |
| FP16 | 71.72 | 80.97 | 71.77 | 74.82 |
| FP6 (e2m3) | 72.09 | 81.43 | 71.55 | 75.02 |
| FP5.3 (e2m3) | 72.27 | 80.36 | 71.97 | **74.87** |
| FP5 (e2m2) | 70.79 | 82.18 | 71.39 | 74.79 |
| FP4.5 (e2m2) | 72.46 | 80.06 | 70.68 | 74.40 |
| FP4.3 (e2m2) | 73.20 | 80.82 | 70.86 | 74.96 |
| FP4.25 (e2m2) | 71.72 | 80.59 | 70.81 | 74.37 |
| FP4 (e2m1) | 71.53 | 77.48 | 69.02 | 72.68 |
| **Llama-3.2-3B-Instruct** | | | | |
| FP16 | 68.21 | 77.41 | 61.73 | 69.12 |
| FP6 (e2m3) | 69.32 | 77.10 | 61.24 | 69.22 |
| FP5.3 (e2m3) | 70.43 | 76.88 | 61.82 | **69.71** |
| FP5 (e2m2) | 67.65 | 75.89 | 60.96 | 68.17 |
| FP4.5 (e2m2) | 71.72 | 74.83 | 59.76 | 68.77 |
| FP4.3 (e2m2) | 68.58 | 73.92 | 59.38 | 67.29 |
| FP4.25 (e2m2) | 69.13 | 73.09 | 59.16 | 67.13 |
| FP4 (e2m1) | 65.25 | 69.22 | 56.74 | 63.74 |
| **Llama-3.1-8B-Instruct** | | | | |
| FP16 | 73.57 | 84.15 | 68.93 | 75.55 |
| FP6 (e2m3) | 74.12 | 84.15 | 68.88 | 75.72 |
| FP5.3 (e2m3) | 74.86 | 84.46 | 68.53 | **75.95** |
| FP5 (e2m2) | 73.94 | 81.88 | 68.19 | 74.67 |
| FP4.5 (e2m2) | 75.60 | 81.73 | 67.51 | 74.95 |
| FP4.3 (e2m2) | 74.49 | 81.12 | 66.80 | 74.14 |
| FP4.25 (e2m2) | 73.75 | 81.27 | 66.97 | 74.00 |
| FP4 (e2m1) | 70.24 | 76.80 | 65.04 | 70.69 |

## 4.2 EFFICIENCY

**Workload** We evaluate the efficiency of our FPX.Y method under linear layers of the Qwen3-4B, Qwen2.5-7B, and Qwen3-32B models. Specifically, we compare our FPX.Y kernels against the baseline W16A16 kernel, W8A16, and TC-FPx kernels to quantify speedups in GEMV-like operations. For each model, we evaluated the latency of each kernel implementation at different batch sizes. We evaluated all kernels on a single GPU with around 22 TFLOPS compute power and 290GB/s memory bandwidth.

**Baseline** For reference, we compare against several widely adopted kernels. The baseline FP16 kernel is the standard cuBLAS (W16A16) implementation of GEMV-like linear layers. For FP6 and FP5 configurations, we use TC-FPx kernels from fp6-llm(commit: 9802c5a)[2]. For the W8A16 comparison, we employ the implementation from TensorRT-LLM (commit: 6837c81), which provides optimized INT8-weight linear kernels widely used in inference frameworks.

**Results** Figure 6 reports the normalized speedups of AMS-based FP5.33 and FP4.25 kernels compared to the FP16 and FP6/FP5 baselines across Qwen3-4B, Qwen2.5-7B, and Qwen3-32B lin-

---

[2]https://github.com/usyd-fsalab/fp6_llm

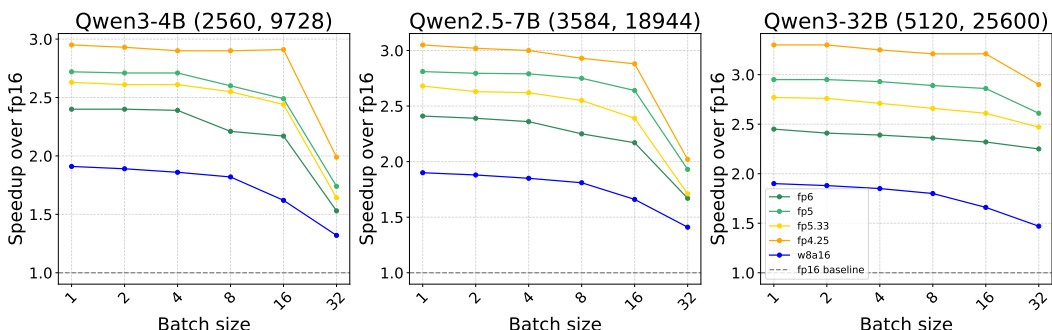

Figure 6: Comparison of kernel speedups versus FP16 baseline across Qwen3-4B, Qwen2.5-7B, and Qwen3-32B MLP-Down linear layers. Dashed line indicates the FP16 baseline (speedup = 1). TC-FPx FP6 (green), TC-FPx FP5 (light green), AMS FP5.33 (gold), AMS FP4.25 (orange), W8A16 (blue).

ear layers. The performance of all GPU kernels tested is normalized using the performance of w16a16 cuBLAS kernels. As shown in Figure 6, AMS kernels consistently outperform all baselines at moderate batch sizes. FP5.33 achieves up to 2.77x/1.46x/1.15x acceleration over cuBLAS(W16A16), TensorRT-LLM(W8A16) and TC-FPx(W6A16), while FP4.25 delivers acceleration up to 3.08x/1.74x/1.11x over cuBLAS(W16A16), TensorRT-LLM(W8A16) and TC-FPx(W5A16).

## 5 RELATED WORKS

Our work falls into the category of weight-only post-training quantization. GPTQ (Frantar et al., 2023) and AWQ (Lin et al., 2024) are widely adopted methods that can perform W4A16 quantization with minimal quality compromise for reasonably-sized LLMs. Research found that just increasing the bit-width beyond 4 bits, such as FP6 (Xia et al., 2024; Wu et al., 2023) and FP5 (Yi et al., 2025) can avoid serious model capability loss for even the simplest RTN quantization. Nevertheless, they still focus on integer bit-width, like 4-bit, 5-bit, 6-bit, 8-bit, etc., while our AMS-Quant can achieve non-integer quantization, like 4.25-bit, 5.33-bit, etc.

We noted that it is possible for GPTQ to realize non-integer quantization by performing per-group quantization and adjusting the grid size. In contrast, AMS-Quant can be applied with any granularity, ranging from simple per-tensor and per-channel to sophisticated per-group quantization. Similar to our works, the line of AQ (Babenko & Lempitsky, 2014) research, including AQLM (Egiazarian et al., 2024) and the subsequent CCQ (Zhou et al., 2025) , can also perform non-integer quantization. However, they have to make use of an auxiliary codebook, which is not as simple and straightforward as our mantissa-bit sharing.

## 6 CONCLUSION

In this work, we proposed AMS-Quant, a novel weight-only floating-point quantization technique to achieve overall non-integer quantization bit-width (e.g., FP5.3), by encoding multiple quantized weight parameters with Adaptive Mantissa-bit Sharing. In this way, we can perform finer-grained optimal bit-width quantization. Applying AMS-Quant to the latest Llama and Qwen series 3∼10B models, we found that FP5.3 models show the same accuracy level as the FP6 models, yet at a higher compression ratio. Moreover, we identified that FP4.25 appears to be the sweet-spot quantization bit-width, which best trades off model quality and inference efficiency.

## LLM USAGE

We used LLMs as tools to make diagrams, organize table data in LATEX syntax, and polish academic writing.

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
