# OpenReview forum: "AMS-Quant: Adaptive Mantissa Sharing for Floating-Point Quantization"
_ICLR.cc/2026/Conference — Submitted to ICLR 2026_

### Official Review · Reviewer_k41G · 2025-10-26

**Soundness:** 4
**Presentation:** 3
**Contribution:** 3
**Rating:** 8
**Confidence:** 3

**Summary:**

This paper proposes AMS-Quant, a novel floating-point quantization method that enables non-integer bit-width representations for LLMs. The method is based on two key components:
- Mantissa-bit Sharing: groups of k quantized weights share their least significant mantissa bit, reducing the effective bit-width.
- Adaptive Searching: an offline optimization that selects the shared bit value minimizing mean squared error across the group.


Authors complement the algorithm with an efficient CUDA kernel implementation that restore quantized weights using bit-level operations.

**Strengths:**

- Idea is novel and timely. While existing works have explored sharing exponent bits or scaling factors, mantissa sharing is surprisingly unexplored.
- The paper is aptly written and clearly organized.
- The proposed adaptive searching for mantissa bits provides an effective mechanism to manage accuracy loss.
- Authors provide substantial experimental results to validate their proposed method.

**Weaknesses:**

- Authors did not report search time for adaptive mantissa sharing algorithm. While it is run offline, it is essential to ascertain the computational cost associated with the search process.
- Non-integer bit-widths based on packing and unpacking bits cannot be integrated with inference frameworks like TensorRT. Likewise, deployment on other hardware devices like TPUs, NPUs etc may need extra effort.

**Questions:**

- What is the time and compute cost for running adaptive mantissa search, especially for large models?
- Can this method be combined with existing methods like GPTQ or AWQ with ease?
- Can this method be extended to activation quantization?

---

> ### Author Response · Authors · 2025-11-28
>
> ### Q1: Offline search time for mantissa sharing algorithm.
>
> We counted the time spent on our AMS\-Quant method across different model sizes, the results are as follows:
>
> |Model|Qwen3-4B|Qwen3-8B|
> |---|---|---|
> |Offline processing time spent|~15min|~30min|
>
> The process is executed on a single 4\-core/2.1GHZ CPU, and as shown in the table above, our method offers negligible overhead offline and can be simply executed without specific calibration. Offline overhead should be better with stronger CPU setups.
>
> ### Q2: Extra effort of deployment under different frameworks and hardware platforms.
>
> Thank you very much for the reviewer’s helpful comment. We would like to clarify that our method was designed with practical hardware considerations in mind.
>
> First, for offline storage, our kernel stores weights in aligned segments \(32\-bit and 16\-bit aligned\). This layout matches the alignment requirements commonly found on GPUs, NPUs, and TPUs, making the format easier to integrate into existing deployment pipelines and Inference framework like TensorRT.
>
> Second, for decoding, we use a small set of bitwise shift\-and\-or operations. These operations are extremely cheap on almost all hardware platforms, so the decoding overhead remains minimal and hardware\-friendly.
>
> Third, besides the CUDA results shown in the paper, we also tested our method on an NPU backend, where we observed similar kernel speedups over the FP16 baseline. This further suggests that the approach can be adapted to different hardware platforms.
>
> We appreciate the reviewer’s comment, as it helps us better clarify the practicality of our method. Thank you again for the valuable feedback.
>
> ### Q3: Can this method be combined with existing methods like GPTQ or AWQ with ease?
>
> Thank you for your advice\! Theoretically it can be combined with outlier\-suppression methods like GPTQ or AWQ, and we did some experiments on AWQ+AMS. We used around 100 calibration data samples to make quantization aware of activation outliers\(AWQ scales are calibrated using the same method as official repos of the AWQ [1] ). Results are as follows.
>
> | **Model**                 | method              | **IFEval** | **GSM8k** | MMLU      | AVG       |
> | ------------------------- | ------------------- | ---------- | --------- | --------- | --------- |
> | **Llama-3.2-3B-Instruct** | FP16                | 68.21      | 77.41     | 61.73     | 69.12     |
> |                           | FP5.3(AMSQuant)     | 70.43      | 76.88     | **61.82** | 69.71     |
> |                           | FP5.3(AMSQuant+AWQ) | **70.79**  | **78.01** | 61.56     | **70.12** |
> | **Llama-3.1-8B-Instruct** | FP16                | 73.57      | 84.15     | **68.93** | 75.55     |
> |                           | FP5.3(AMSQuant)     | **74.86**  | **84.46** | 68.53     | **75.95** |
> |                           | FP5.3(AMSQuant+AWQ) | 73.38      | 84.31     | 68.55     | 75.41     |
>
> As shown in the tables above, AWQ+AMS\-Quant did not outperform pure AMS\-Quant under every setup. We think the rationale lies under that the AMS\-format's dynamic range is better than INT\-format, which leads to a better endurance for outliers in weight tensors. Consequently, combining with AWQ does not bring consistent additional accuracy improvement. Thank you for your advice on combining existing methods and our method, and further study will delve into this aspect.
>
> ### Q4: Can this method be extended to activation quantization?
>
> Thank you for your advice. We did extend AMS\-Quant to activation quantization, and here is the result of wXa8 quantization.
>
> | **Model**              | method               | **IFEval** | **GSM8k** |
> | ---------------------- | -------------------- | ---------- | --------- |
> | Qwen3-4B-Instruct-2507 | w16a16               | 80.96      | 89.69     |
> | | w8a8(SmoothQuant)    | 82.26      | 88.86     |
> | | **w5.3a8(AMSQuant)** | 80.96      | 89.46     |
> | Llama-3.2_3B-Instruct  | w16a16               | 68.21      | 77.41     |
> | | w8a8(SmoothQuant)    | 68.95      | 77.86     |
> | | **w5.3a8(AMSQuant)** | 68.95      | 76.16     |
> | Qwen2.5-7B-Instruct    | w16a16               | 71.72      | 80.59     |
> | | w8a8(SmoothQuant)    | 71.16      | 79.98     |
> | | **w5.3a8(AMSQuant)** | 69.13      | 81.12     |
> | Llama-3.1-8B-Instruct  | w16a16               | 73.57      | 84.15     |
> | | w8a8(SmoothQuant)    | 72.46      | 84.31     |
> | | **w5.3a8(AMSQuant)** | 75.60       | 84.91      |
>
> **The result shows that our method can be integrated with the activation quantization method without significant accuracy loss.** It shows that the AMS\-Quant can be well extended to activation quantization.
>
> We hope these responses fully address your questions. Thank you again for your constructive feedback\! We welcome any further discussion or suggestions.
>
> ---
>
> [1] https://github.com/mit-han-lab/llm-awq/tree/main/awq

---

### Official Review · Reviewer_5aq1 · 2025-10-30

**Soundness:** 2
**Presentation:** 3
**Contribution:** 2
**Rating:** 4
**Confidence:** 4

**Summary:**

This paper presents AMS-Quant, a weight-only floating-point quantization framework for LLMs. It includes two major techniques, i.e., mantissa-bit sharing to boost compression and adaptive searching to reduce quantization error. A custom GPU kernel is developed to realize practical speedups. Experiments show that AMS-Quant achieves these gains with only minimal accuracy degradation.

**Strengths:**

1. The paper is well-written and the core ideas are clear to understand.
2. The major contributions of this quantization framework are mantissa sharing and adaptive searching, which are both sound for improved floating-point quantization.
3. The experimental results not only show the quantization accuracy but also the actual acceleration enabled by the proposed methods.

**Weaknesses:**

1. This work is for weight-only quantization, if it can be further extended to an activation and weight quantization, more efficiency gains should be obtained.
2. The mantissa-sharing operation is analogous to block floating-point quantization and micro-scaling formats (e.g., MXINT, MXFP), which share exponents instead. Additional experiments should be conducted to more thoroughly validate the superiority of the proposed methods.
3. Mantissa sharing is applied along the input-channel dimension of the weight tensor. However, this choice is less cache- and memory-friendly than sharing along the output-channel dimension. While the paper justifies the design by noting that activation outliers are typically aligned with input channels, it may be worthwhile to incorporate a small calibration set into the mantissa-sharing and adaptive search procedure to reduce quantization loss, while enabling mantissa sharing along the output-channel dimension for more efficient memory access.

Overall, this is a good-quality paper and I will raise my score if my concerns are well addressed.

**Questions:**

See weaknesses.

---

> ### Author Response · Authors · 2025-11-28
>
> ### Q1: Further extended to an activation and weight quantization.
>
> Thank you for your advice on extending our methods to activation and weight quantizatio. Results of wXa8 quantized models are as follows. The w5.33a8 experiment is set up with RTN FP8 activation quantization, while w8a8 SmoothQuant weights are directly downloaded from [HuggingFace official repos](https://huggingface.co/RedHatAI).
>
> | **Model**              | method               | **IFEval** | **GSM8k** |
> | ---------------------- | -------------------- | ---------- | --------- |
> | Qwen3-4B-Instruct-2507 | w16a16               | 80.96      | 89.69     |
> | | w8a8(SmoothQuant)    | 82.26      | 88.86     |
> | | **w5.3a8(AMSQuant)** | 80.96      | 89.46     |
> | Llama-3.2_3B-Instruct  | w16a16               | 68.21      | 77.41     |
> | | w8a8(SmoothQuant)    | 68.95      | 77.86     |
> | | **w5.3a8(AMSQuant)** | 68.95      | 76.16     |
> | Qwen2.5-7B-Instruct    | w16a16               | 71.72      | 80.59     |
> | | w8a8(SmoothQuant)    | 71.16      | 79.98     |
> | | **w5.3a8(AMSQuant)** | 69.13      | 81.12     |
> | Llama-3.1-8B-Instruct  | w16a16               | 73.57      | 84.15     |
> | | w8a8(SmoothQuant)    | 72.46      | 84.31     |
> | | **w5.3a8(AMSQuant)** | 75.6       | 84.9      |
>
> The result shows that **our method can be integrated with the activation quantization method without significant accuracy loss compared with the FP16 original model**. Further study will dig into finer\-grained activation quantization and lower bits for activations.
>
> ### Q2: Comparison to micro\-scaling formats such as \(MXFP/MXINT\).
>
> Here are the results of micro\-scaling formats' models for weight\-only quantization tests:
>
> | **Model**                  | method              | **IFEval** | **GSM8k** | MMLU      | AVG       |
> | -------------------------- | ------------------- | ---------- | --------- | --------- | --------- |
> | Meta-Llama-3.1-8B-Instruct | *FP16*              | *73.57*    | *84.15*   | *68.93*   | *75.55*   |
> | | MXFP4               | 66.91      | 69.14     | 61.72     | 65.92     |
> | | **AMSQuant(fp4.5)** | **75.6**   | 81.73     | 67.51     | 74.95     |
> | | **AMSQuant(fp5.3)** | 74.86      | **84.46** | **68.53** | **75.95** |
> | Meta-Llama-3.2-3B-Instruct | *FP16*              | *68.21*    | *77.41*   | *61.73*   | *69.12*   |
> | | MXFP4               | 63.77      | 69.52     | 57.9      | 63.73     |
> | | **AMSQuant(fp4.5)** | **71.72**  | 74.83     | 59.76     | 68.77     |
> | | **AMSQuant(fp5.3)** | 70.43      | **76.88** | **61.82** | **69.71** |
>
> \(For the MXFP4 method, we use E8M0 per\-block scales with block size of 32\)
>
> As shown in the tables above, **our method consistently outperforms micro-scaling format quantization**. During our accuracy experiments, we found that the dynamic range of weight tensors in LLMs is far smaller than activations. *Hence, it is be a better choice to assign more bits to mantissa that control granularity instead of exponents which contribute to dynamic ranges.*
>
> Similar conclusions can be found in a paper [1] comparing FP4/MXFP4/NVFP4 that granularity is more important than dynamic range when it comes to weight-only quantization.
>
> ### Q3: Channels to choose in the mantissa sharing.
>
> Thank you for your suggestion of applying mantissa sharing on the output channel, and we agree that it will benefit at kernel level in CUDA environments. The accuracy results of "output\-channel\-wise mantissa sharing" are as below. \(Following the setting of AWQ, we used around 100 calibration data samples to make quantization aware of activation outliers.\)
>
>
> | **Model**                 | method              | **IFEval** | **GSM8k** | MMLU        | AVG       |
> | ------------------------- | ------------------- | ---------- | --------- | ----------- | --------- |
> | **Llama-3.2-3B-Instruct** | *FP16*              | *68.21*    | *77.41*   | *61.73*     | *69.12*   |
> | | FP5.3(AMSQuant)     | 70.43      | 76.88     | **61.82**   | 69.71     |
> | | FP5.3(AMSQuant+AWQ) | **70.79**  | **78.01** | 61.56       | **70.12** |
> | **Llama-3.1-8B-Instruct** | *FP16*              | *73.57*    | *84.15*   | ***68.93*** | *75.55*   |
> | | FP5.3(AMSQuant)     | **74.86**  | **84.46** | 68.53       | **75.95** |
> | | FP5.3(AMSQuant+AWQ) | 73.38      | 84.31     | 68.55       | 75.41     |
>
> As shown in the tables above, "output channel mantissa\-sharing" combined with "Activation\-aware calibration" did not outperform "input channel mantissa \-sharing". Furthermore, the current design naturally takes activation outliers into consideration, while saving the overhead of calibration.
>
> ---
>
> We hope these responses fully address your questions. Thank you again for your constructive feedback\! We welcome any further discussion or suggestions.
>
> [1] Egiazarian, Vage, et al. "Bridging the Gap Between Promise and Performance for Microscaling FP4 Quantization."

---

### Official Review · Reviewer_j2b7 · 2025-10-31

**Soundness:** 1
**Presentation:** 2
**Contribution:** 1
**Rating:** 0
**Confidence:** 4

**Summary:**

This paper focuses on weight-only quantization by leveraging mantissa-bit sharing, resulting in non-integer bit widths for individual weight elements. It also proposes an adaptive search strategy to determine the optimal shared mantissa value. Based on these techniques and a bit-slicing packing strategy, the authors implement CUDA linear kernels and demonstrate improved latency through more efficient memory access.

**Strengths:**

They implement CUDA kernels using an efficient restoration algorithm based on the bit-slicing technique.

**Weaknesses:**

A major limitation of this paper is the lack of comparison with existing methods. Numerous prior works achieve higher (even 2-bit in QuIP or AQLM) compression rates while maintaining comparable accuracy. Without extensive benchmarking against these methods, it is difficult to evaluate the advantages of the proposed strategy. Furthermore, many weight-activation quantization (e.g., Quarot) approaches exist that effectively address KV cache and high-precision computation overhead in attention mechanisms, which makes the proposed method less compelling.

Additionally, the rationale for mantissa-bit sharing is not clearly justified. The method - merely sharing the least significant bits of the mantissa and selecting the optimal value via brute-force search - appears simplistic and does not convincingly constitute a novel contribution. From a practical perspective, applying techniques that reduce additional 1 or 2 bits more substantially might be more impactful than the minor gains achieved through mantissa sharing.

Regarding the packaging and restoration techniques, the approach shows limited novelty compared to the TC-FPx framework and appears largely as an engineering implementation. Finally, the figures do not effectively convey the core concepts and seem to underutilize the available space, reducing their clarity and impact.

**Questions:**

What are the advantages of using non-integer bit widths and the mantissa-sharing strategy compared to existing quantization methods? Please clarify how this approach provides benefits in terms of accuracy, compression, or computational efficiency relative to prior work.

---

> ### Author Response · Authors · 2025-11-28
>
> ### Q1: Lack of comparison with existing methods.
> Thank you for your suggestion on providing more baselines, and correspondingly, we compared our method with AWQ, GPTQ and NF4  methods across different models and datasets (the best results are highlighted).
>
>
> | Model                              | Method              | IFEval    | GSM8k     | MMLU      | AVG       |
> | ---------------------------------- | ------------------- | --------- | --------- | --------- | --------- |
> | **Qwen3\-4B\-2507\-Instruct**      | *FP16*              | *80.96*   | *89.69*   | *72.45*   | *81.03*   |
> |                                    | AWQ                 | 79.85     | 86.5      | 70.84     | 79.06     |
> |                                    | GPTQ                | 80.41     | **89.92** | 71.01     | 80.45     |
> |                                    | NF4                 | 79.85     | 89.08     | 71.33     | 80.09     |
> |                                    | **AMSQuant(fp4.5)** | 80.04     | 88.7      | 71.49     | 80.08     |
> |                                    | **AMSQuant(fp5.3)** | **81.7**  | 88.86     | **71.74** | **80.77** |
> | **Meta\-Llama\-3.1\-8B\-Instruct** | *FP16*              | *73.57*   | *84.15*   | *68.93*   | *75.55*   |
> |                                    | AWQ                 | 72.83     | 83.78     | 66.98     | 74.53     |
> |                                    | GPTQ                | 73.75     | 82.03     | 67.31     | 74.36     |
> |                                    | NF4                 | 72.83     | 81.12     | 66.16     | 73.37     |
> |                                    | **AMSQuant(fp4.5)** | **75.60** | 81.73     | 67.51     | 74.95     |
> |                                    | **AMSQuant(fp5.3)** | 74.86     | **84.46** | **68.53** | **75.95** |
> | **Meta\-Llama\-3.2\-3B\-Instruct** | *FP16*              | *68.21*   | *77.41*   | *61.73*   | *69.12*   |
> |                                    | AWQ                 | 68.95     | 76.88     | 60.57     | 68.80     |
> |                                    | GPTQ                | 67.1      | 71.11     | 57.94     | 65.38     |
> |                                    | NF4                 | 69.5      | 73.84     | 59.94     | 67.76     |
> |                                    | **AMSQuant(fp4.5)** | **71.72** | 74.83     | 59.76     | 68.77     |
> |                                    | **AMSQuant(fp5.3)** | 70.43     | **76.88** | **61.82** | **69.71** |
> | **Qwen2.5\-7B\-Instruct**          | *FP16*              | *71.72*   | *80.97*   | *71.77*   | *74.82*   |
> |                                    | AWQ                 | 70.06     | **81.05** | 70.91     | 74.01     |
> |                                    | GPTQ\-Int4          | 70.79     | 80.97     | 70.18     | 73.98     |
> |                                    | NF4                 | 69.13     | 78.24     | 69.47     | 72.28     |
> |                                    | **AMSQuant(fp4.5)** | **72.46** | 80.06     | 70.68     | 74.40     |
> |                                    | **AMSQuant(fp5.3)** | 72.27     | 80.36     | **71.97** | **74.87** |
>
>
>
> As shown in the tables above, our method continuously outperformed other frequently\-used weight\-only quantization methods, and is nearly lossless compared to the original FP16 results.

---

> ### Author Response · Authors · 2025-11-28
>
> ### Q2: Comparison to extreme low bit quantization, such as 2 or 3 bits.
>
> The proposed AMS\-Quant aims to approach the lower\-bound bit\-width of the model while keeping the full accuracy level instead of pursuing extreme efficiency at the cost of  sacrificing model accuracy. According to the accuracy results provided in official papers of QuIP[1] and AQLM[2], when it comes to extremely low bits quantization under 3bits with model size of 4\-13B, accuracy loss was around 2 to 5 points on average. In our paper, we showed that our method can maintain nearly lossless accuracy.
>
> From a practical perspective, many realistic cases are sensitive to accuracy loss, such as autonomous driving scenarios, under which the accuracy loss of extremely low bit quantization can be unacceptable. Furthermore, the accuracy between a 14B model and a 32B model or a 4B model and a 8B moodel are merely 5 points on average [3] [4], so we consider that such 2 to 5 points accuracy loss would hurt the competence of the original model. In conclusion, it is important to pursue "nearly lossless" low\-bit quantization, which is exactly one of our method's motivation.
>
> ### Q3: QuaRot, integration with KVCache and attention compression.
>
> Our method is orthogonal to activation compression and KVCache compression methods, such as AWQ and QuaRot, so it can be combined with these activation and KVCache compression methods. Thank you for your advice on extending our method, and we will take this into consideration in our future studies.
>
> ### Q4: Rationale of mantissa sharing.
>
> As is stated in section 3.1, the optimization objective of mantissa sharing is the same as round\-to\-nearest quantization, that is, to minimize the reconstruction error of quantized weight.
>
> We used MSE \(Mean Squared Error\) as the objective function, and the results demonstrated in this paper have shown that it is a simple but effective criterion of reconstruction error. Thank you for your advice on trying a different approach to find the shared bit.
>
> ### Q5: Kernel novelty compared to TC\-FPx kernels.
>
> Thank you for your question. As we stated in Section 3.2, our kernel implementation is based on the TC\-FPx framework. We shall highlight that **the novelty of this paper does not lie in the kernel, but our adaptive mantissa sharing method to approach the minimum required bit\-width of a model at the full original accuracy level.**
>
> ### Q6: Clarity of figures of kernel efficiency.
>
> Thank you for your opinion on the figures of our paper, and we will polish our main figure in the revision.
>
> ---
>
> We hope these responses fully address your questions. Thank you again for your constructive feedback\! We welcome any further discussion or suggestions.
>
>
>
> [1] Tseng et al. "QuIP\#: Even Better LLM Quantization with Hadamard Incoherence and Lattice Codebooks."
>
> [2] Egiazarian et al. "Extreme Compression of Large Language Models via Additive Quantization."
>
> [3] Yang, An, et al. "Qwen2.5 Technical Report"
>
> [4] Yang, An, et al. "Qwen3 technical report."

---

### Official Review · Reviewer_VgPA · 2025-10-31

**Soundness:** 3
**Presentation:** 2
**Contribution:** 2
**Rating:** 4
**Confidence:** 3

**Summary:**

The authors propose mantissa-bit sharing, sharing the least significant mantissa bit across groups of k quantized weights, resulting in formats like FP5.33-e2m3 and FP4.25-e2m2 (1:3 / 1:4 sharing). Per group 1-bit search minimizies the quantization MSE. Further, they implement bit-packing + register level ops for restoring FP16 for CUDA linear kernels. Speedup over FP16 kernels are demonstrated. Finally, their FP5.3(e2m3) hits the sweet-spot across IFEval GSM8k and MMLU on 3 models.

**Strengths:**

- Simple to implement and leverage
- design is sound, kernel also obtains a speedup in memory-bound settings at the kernel-level.

**Weaknesses:**

- baselines seem to be limited, AWQ.GPTQ, NF4 should be discussed.
- Are the speedups only kernel-level? it is very important to see full decode latency,  specifically is that a speedup for the model, or for just the kernel in Figure 6? I think its the latter so it might be better to not label them as real model speedups.

**Questions:**

- Beyond notes on the weaknesses, it would be interesting to see an ablation of increasing k.

---

> ### Author Response · Authors · 2025-11-28
>
> ### Q1: Baselines seem to be limited, AWQ, GPTQ, NF4 should be discussed.
>
> Thank you for your suggestion on providing more baselines, and correspondingly, we compared our method with AWQ, GPTQ and NF4  methods across different models and datasets (the best results are highlighted).
>
> | Model                              | Method              | IFEval    | GSM8k     | MMLU      | AVG       |
> | ---------------------------------- | ------------------- | --------- | --------- | --------- | --------- |
> | **Qwen3\-4B\-2507\-Instruct**      | *FP16*              | *80.96*   | *89.69*   | *72.45*   | *81.03*   |
> |                                    | AWQ                 | 79.85     | 86.5      | 70.84     | 79.06     |
> |                                    | GPTQ                | 80.41     | **89.92** | 71.01     | 80.45     |
> |                                    | NF4                 | 79.85     | 89.08     | 71.33     | 80.09     |
> |                                    | **AMSQuant(fp4.5)** | 80.04     | 88.7      | 71.49     | 80.08     |
> |                                    | **AMSQuant(fp5.3)** | **81.7**  | 88.86     | **71.74** | **80.77** |
> | **Meta\-Llama\-3.1\-8B\-Instruct** | *FP16*              | *73.57*   | *84.15*   | *68.93*   | *75.55*   |
> |                                    | AWQ                 | 72.83     | 83.78     | 66.98     | 74.53     |
> |                                    | GPTQ                | 73.75     | 82.03     | 67.31     | 74.36     |
> |                                    | NF4                 | 72.83     | 81.12     | 66.16     | 73.37     |
> |                                    | **AMSQuant(fp4.5)** | **75.60** | 81.73     | 67.51     | 74.95     |
> |                                    | **AMSQuant(fp5.3)** | 74.86     | **84.46** | **68.53** | **75.95** |
> | **Meta\-Llama\-3.2\-3B\-Instruct** | *FP16*              | *68.21*   | *77.41*   | *61.73*   | *69.12*   |
> |                                    | AWQ                 | 68.95     | 76.88     | 60.57     | 68.80     |
> |                                    | GPTQ                | 67.1      | 71.11     | 57.94     | 65.38     |
> |                                    | NF4                 | 69.5      | 73.84     | 59.94     | 67.76     |
> |                                    | **AMSQuant(fp4.5)** | **71.72** | 74.83     | 59.76     | 68.77     |
> |                                    | **AMSQuant(fp5.3)** | 70.43     | **76.88** | **61.82** | **69.71** |
> | **Qwen2.5\-7B\-Instruct**          | *FP16*              | *71.72*   | *80.97*   | *71.77*   | *74.82*   |
> |                                    | AWQ                 | 70.06     | **81.05** | 70.91     | 74.01     |
> |                                    | GPTQ\-Int4          | 70.79     | 80.97     | 70.18     | 73.98     |
> |                                    | NF4                 | 69.13     | 78.24     | 69.47     | 72.28     |
> |                                    | **AMSQuant(fp4.5)** | **72.46** | 80.06     | 70.68     | 74.40     |
> |                                    | **AMSQuant(fp5.3)** | 72.27     | 80.36     | **71.97** | **74.87** |
>
> \(AWQ and GPTQ weights are directly downloaded from official repos on HuggingFace, while NF4 weights are reproduced based on the source code of Qlora[1], and block size is set to 64 for all models\). As shown in the table above,
> 1. The FP5.33 quantization consistently outperforms the other quantization baselines, with the average score being roughly one point higher. Its performance is also nearly lossless compared to the original full\-precision model.
> 2. The FP4.5 quantization achieves accuracy that is generally comparable to or better than the other baseline methods, with a negligible accuracy loss of no more than 0.5%
>
> This leads to the conclusion that **our method can better maintain the original model's accuracy than other frequently-used weight-only quantization methods**.
>
> [1] https://github.com/artidoro/qlora

---

> ### Author Response · Authors · 2025-11-28
>
> ### Q2: Are the speedups only kernel\-level?
> Thank you for pointing out the potential ambiguity of speedup conclusions, and exactly as you assumed, it's kernel-level speedups in the original paper. To avoid confusion, we will revise the paper in future versions.
>
> What’s more, we made some effort to integrate our kernel into inference frameworks to validate E2E speedups: Due to hardware limitations, we compress model size and approximate E2E performance using models with fewer layers, whilst maintaining the same shape for each layer. We focused on memory\-bound scenarios as it's the main purpose of doing weight\-only quantization. The efficiency results of Qwen3-32B and Qwen3-8B of different batch sizes are as follows.
>
>
> |   Qwen3 - 8B       |   fp6    |  fp5.33  |   fp5    |  fp4.25  |
> |----------|:--------:|:--------:|:--------:|:--------:|
> | bs=1     | $1.54 \times$ | $1.61 \times$ | $1.64 \times$ | $1.70 \times$ |
> | bs=4     | $1.32 \times$ | $1.41 \times$ | $1.42 \times$ | $1.46 \times$ |
> | bs=8     | $1.14 \times$ | $1.17 \times$ | $1.19 \times$ | $1.22 \times$ |
>
> |     Qwen3 - 32B      |   fp6    |  fp5.3   |   fp5    |  fp4.25  |
> |----------|:--------:|:--------:|:--------:|:--------:|
> | bs=1     | $1.93 \times$ | $2.04 \times$ | $2.09 \times$ | $2.21 \times$ |
> | bs=4     | $1.80 \times$ | $1.89 \times$ | $1.93 \times$ | $2.01 \times$ |
> | bs=8     | $1.55 \times$ | $1.67 \times$ | $1.69 \times$ | $1.74 \times$ |
>
>
>
> (Numbers in the table are speedup ratios compared to the FP16-format model) As shown in the table above, kernel-level speedup can be extended to model\-level, as in memory-bound scenarios, it is the data copy of weights that takes up most of the inference time.
>
> ### Q3: Ablation study of increasing K.
> Thank you for your advice on experiments on increasing K. We gradually increased K and reduced the average bit width between 4\-5 bits. The model's average accuracy showed a slow downward trend, while it consistently exceeded the accuracy of the FP4 model, which again shows the effectiveness of the proposed AMS\-Quant. Results of ablation study are as follows.
>
> | Llama-3.1-8B-Instruct | **K** | **IFEval (Strict)** | **GSM8k** | **MMLU** | **AVG** |
> | --------------------- | ----- | ------------------- | --------- | -------- | ------- |
> | fp16                  | /     | 73.57               | 84.15     | 68.93    | 75.55   |
> | fp4                   | /     | 70.24               | 76.8      | 65.04    | 70.69   |
> | fp4.1                 | 10    | 74.49               | 80.29     | 66.51    | 73.76   |
> | fp4.125               | 8     | 72.54               | 80.82     | 66.31    | 73.22   |
> | fp4.167               | 6     | 75.97               | 80.06     | 66.21    | 74.08   |
> | fp4.25                | 4     | 73.75               | 81.27     | 66.97    | 74.00   |
> | fp4.33                | 3     | 74.49               | 81.12     | 66.8     | 74.14   |
> | fp4.5                 | 2     | 75.6                | 81.73     | 67.51    | 74.95   |
>
> | **Llama-3.2-3B-Instruct** | **K** | **IFEval (Strict)** | **GSM8k** | **MMLU** | **AVG** |
> | ------------------------- | ----- | ------------------- | --------- | -------- | ------- |
> | fp16                      | /     | 68.21               | 77.41     | 61.73    | 69.12   |
> | fp4                       | /     | 65.25               | 69.22     | 56.74    | 63.74   |
> | fp4.1                     | 10    | 65.99               | 70.51     | 57.87    | 64.79   |
> | fp4.125                   | 8     | 65.62               | 70.58     | 58.12    | 64.77   |
> | fp4.167                   | 6     | 68.39               | 71.34     | 58.35    | 66.03   |
> | fp4.25                    | 4     | 69.13               | 73.09     | 59.16    | 67.13   |
> | fp4.33                    | 3     | 68.58               | 73.92     | 59.38    | 67.29   |
> | fp4.5                     | 2     | 71.72               | 74.83     | 59.76    | 68.77   |
>
> We hope these responses fully addressed your questions. Thank you again for your constructive feedback!

---

### Author Response · Authors · 2025-11-28

We thank all reviewers for their constructive feedback and thoughtful questions. Across the reviews, several common strengths of our work were highlighted, including:
- **The novelty of our adaptive mantissa sharing mechanism design is acknowledged.**
- The clear motivation around improving low-bit floating-point quantization in both aspects of accuracy and efficiency.
- The close algorithm–system co-design of our quantization and kernel optimizations.
We are grateful for these recognitions. Below, we summarize the major revisions and clarifications made in response to reviewers’ comments. All primary conclusions of the paper remain unchanged.

### Expanded baseline comparisons (AWQ, GPTQ, NF4).

Reviewers VgPA suggested including more baselines—particularly AWQ, GPTQ, and NF4. We have added comprehensive comparisons across various models (see response to Reviewer VgPA and j2b7). The results our AMS-Quant can outperform the all the added mentioned state-of-the-art methods.

###  Combination of existing methods with our method.

We combined activation-aware methods(AWQ) with our method, using the same method to calibrate AWQ-scales according to the AWQ official repo to ensure consistency. AmsQuant results have been updated (see response to Reviewer k41G).

### Extension to activation quantization (w5.3a8).

We combined the activation quantization method with our weight-only quantization method across Llama family Models and compared accuracy results to w8a8 smoothquant models and original models(see response to Reviewer 5aq1 and k41G). Results show that our method can extend to activation quantization with no significant accuracy loss.

### Ablation study on increasing K.

Reviewer VgPA suggests that we make an ablation study of increasing K (see response to Reviewer VgPA). Through the ablation study, fp4.x of our method continuously outperformed fp4 format with a moderate degradation of accuracy when K increases, which shows the effectiveness of our method.

---

### Meta-Review · Area_Chair_75wD · 2025-12-22

**Summary:**

This paper presents AMS-Quant, a floating-point quantization framework for LLMs. It includes two major techniques, i.e., mantissa-bit sharing to boost compression and adaptive search to reduce quantization error. A custom GPU kernel is developed to realize practical speedups. Experiments show that AMS-Quant achieves these gains with only minimal accuracy degradation over FP16 inference. Four reviewers provided their reviews, but none responded to the rebuttal in a timely manner. The authors supplied more experimental results comparing more baselines, combination with existing methods, extension to activation quantization, and an ablation on k. These new results strengthen the paper.

However, the AC does not think the authors have successfully addressed all proposed concerns. Specifically, the AC agrees with reviewer j2b7's concerns about low-bit baselines, and the rationale for mantissa-bit sharing is not clearly justified. Therefore, the AC is slightly inclined to reject this submission and encourage the authors to include these improvements for a future submission.

**Reviewer Concerns:**

1. Reviewer VgPA (mostly addressed).
2. Reviewer j2b7 (minorly addressed). Unsolved concerns: 1)comparison with 2-3 bits baselines 2) the rationale for mantissa-bit sharing is not clearly justified and limited novelty 3) the figures do not effectively convey the core concepts
3. Reviewer 5aq1(partly addressed). The AC believe it's better to include the real inference latency to Q2, which can measure the overehead of applying mantissa sharing on the input channel
4. Reviewer k41G (mostly addressed)

**Reviewer Scores:**

1. Reviewer VgPA would keep 4 or increase the score to 6.
2. Reviewer j2b7 would keep the score as 0 or at most, increase it to 2.
3. Reviewer 5aq1 would keep 4 or increase the score to 6.
4. Reviewer k41G  would keep the score as 8.

---

### Decision · Program_Chairs · 2026-01-26

Reject